# Amharic Text Normalization with Sequence-to-Sequence Models

## 1. Introduction

Text normalization or the transformation of words from the written to the spoken form is an important problem. It is also quite a complex problem, due to the range of different kinds of non-standard words (NSW's), the special processing required for each case, and the propensity for ambiguity among NSW's as a class. Unfortunately, text normalization is not a problem that has received a great deal of attention.

## 2. Method

We have collected data that includes many types of non-standard words from different Amharic news Media and websites, FBC (Fana Broadcasting Corporate) more than eighty percent, VOA (Voice of America) and BBC (British Broadcasting corporate) with four types of news domain, including political, economic, sport and health, which contributes to the inclusion of most of Amharic NSWs.

After data gathering and preprocessing, the processed text is an input for our general architecture first component, tokenization, a process of segmentation of collected document into sentence-level and then to word level. Later every token has to pass through a token identification process that identifies its token class type, from which we are going to determine its expansion form for training data preparation.

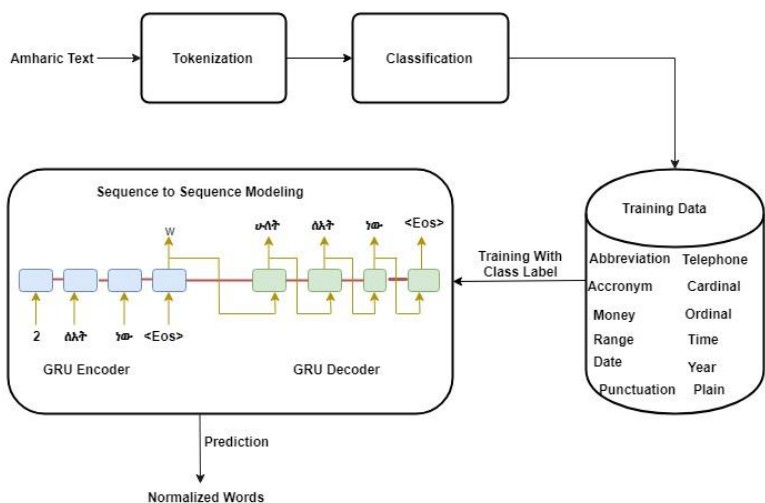

Figure 1: A Framework for Amharic Text Normalization

Detecting the class of the token or the taxonomy is the key part of this task. Once we have determined the class of a token correctly, we expand it based on our rule-based normalizer accordingly. The usage of a token in a sentence determines its class. To determine the class of the token in focus, the surrounding tokens play an important role.

Finally, the training data is the input for our architecture last component, the sequence-to-sequence architecture, which model the whole task as one where we map a sequence of input characters to a sequence of output words.

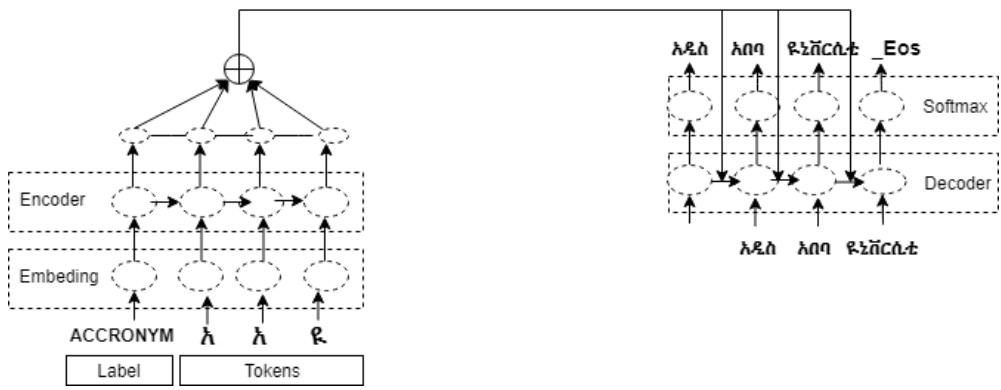

Figure 2: The Sequence-to-Sequence Architecture

Our architecture performs at a character-level input sequence and word-level output sequence. We used an embedding layer, Bidirectional 3-layer Gated Recurrent Units (GRU) encoder that reads input tokens, and a one-layer GRU decoder that produces word sequences. These are the translation for the architecture inputs and outputs, "ኢ" for "A", "ዩ" for "U", "አዲስ" for "Addis", "አበባ" for "Ababa" and "ዩኒቨርሲቲ" for "University".

## 3. Experiment and Results

We use bidirectional GRU with the size of 250 hidden units both for encoding and decoding layers. The training takes 7 days on the whole 200 thousand sentences training data on 8GB RAM with 2.5 GHz intel core i5 Laptop. Across all experiments, to reduce the input dimension, we choose character-based solution over word-based.

| Class | Word Count | Accuracy |
|---|---|---|
| PLAIN | 96,101 | 0.996 |
| PUNC | 7,629 | 1.000 |
| ACRYM | 1,320 | 0.998 |
| ABR | 1,402 | 0.986 |
| CARD | 3,430 | 0.926 |
| ORD | 1,275 | 0.969 |
| MONEY | 1,104 | 0.872 |
| DATE | 4,509 | 0.899 |
| YEAR | 2,880 | 0.859 |
| TIME | 4,690 | 0.861 |
| TELE | 1,606 | 0.895 |
| RANGE | 1,208 | 0.891 |

Table 1: Accuracy of Experiment 4

## 4. Conclusion

In this work, based on our final experiment results, we found 94.8 percent accuracy which is promising, considering that we trained on a small annotated dataset. Besides, on top of the corpus we prepared, we believe the performance of the model can be further improved by designing a more balanced training dataset.

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
