# OpenReview forum: "Amharic Text Normalization with Sequence-to-Sequence Models"
_ICLR.cc/2020/Conference — Reject_

### Official Review · AnonReviewer3 · 2019-10-25
**Official Blind Review #3**

**Rating:** 1

**Review:**

This paper addresses the text normalization problem, where  the special processing is required for different kinds of non-standard words (NSW’s).

Dataset:  a new dataset is collected, including many types of non-standard words from different Amharic news Media
and websites, FBC more than eighty percent, VOA and BBC.

Model: Bidirectional GRU with the size of 250 hidden units both are used for encoding and decoding layers.

This paper is not ready to publish. Please consider to complete the project, polish the writing, and submit to a different venue.



**Experience Assessment:**

I have published one or two papers in this area.

**Review Assessment: Checking Correctness Of Derivations And Theory:**

I assessed the sensibility of the derivations and theory.

**Review Assessment: Checking Correctness Of Experiments:**

I assessed the sensibility of the experiments.

**Review Assessment: Thoroughness In Paper Reading:**

I read the paper at least twice and used my best judgement in assessing the paper.

---

### Official Review · AnonReviewer1 · 2019-11-03
**Official Blind Review #1**

**Rating:** 1

**Review:**

Text normalization or the transformation of words from the written to the spoken form is an important and realistic question in natural language processing. This paper aims to use sequence-to-sequence models to perform text normalization.

However, this paper does not use the official template and the content is too short to be a conference paper.
I suggested resubmitting to another (NLP) conference after extending the content with detailed description for the model and the method, and conducting more experiments on public acceptable benchmarks.

**Experience Assessment:**

I have read many papers in this area.

**Review Assessment: Checking Correctness Of Derivations And Theory:**

I assessed the sensibility of the derivations and theory.

**Review Assessment: Checking Correctness Of Experiments:**

I did not assess the experiments.

**Review Assessment: Thoroughness In Paper Reading:**

I made a quick assessment of this paper.

---

### Official Review · AnonReviewer4 · 2019-11-08
**Official Blind Review #4**

**Rating:** 1

**Review:**

The paper describes a method for word normalization of Amharic text using a word classification system followed by a character-based GRU attentive encoder-decoder model.

The paper is very short and lacks many important details, such as where the data is collected from, how it is processed and split into training and evaluation sets, and how the initial token classification is performed. The paper also doesn't adhere to the conference paper template, which is grounds for desk rejection.

The authors should revise the paper with this information and consider submitting to a different venue, as the task considered, while interesting, seems far from the core focus of ICLR.


**Experience Assessment:**

I have read many papers in this area.

**Review Assessment: Checking Correctness Of Derivations And Theory:**

N/A

**Review Assessment: Checking Correctness Of Experiments:**

I assessed the sensibility of the experiments.

**Review Assessment: Thoroughness In Paper Reading:**

I read the paper thoroughly.

---

### Decision · Program_Chairs · 2019-12-19

**Decision:**

Reject

**Comment:**

The paper proposes a text normalisation model for Amharic text. The model uses word classification, followed by a character-based GRU attentive encoder-decoder model. The paper is very short and does not present reproducible experiments. It also does not conform to the style guidelines of the conference. There has been no discussion of this paper beyond the initial reviews, all of which reject it with a score of 1. It is not ready to publish and the authors should consider a more NLP focussed venue for future research of this kind.